# Study on Quality Changes of Kelp Gel Edible Granules during Storage

**DOI:** 10.3390/foods13142267

**Published:** 2024-07-18

**Authors:** Tingru Chen, Ying Li, Yin Wang, Jicheng Chen, Lin’ao Fan, Zhiyu Liu

**Affiliations:** 1Key Laboratory of Cultivation and High-Value Utilization of Marine Organisms in Fujian Province, National and Local Joint Engineering Research Center of Marine Biological Seed Industry Technology, Fisheries Research Institute of Fujian, Xiamen 361013, China; ctr1003533034@163.com (T.C.); wangyin_83@163.com (Y.W.); 2College of Food Science, Fujian Agriculture and Forestry University, Fuzhou 350002, China; 15180261868@163.com (Y.L.); newtaicjc@163.com (J.C.); 15359131221@163.com (L.F.)

**Keywords:** kelp, gel, edible granules, storage, quality changes

## Abstract

The kelp gel edible granules developed utilizing the gel properties of alginate are prone to quality deterioration if improperly stored during the storage process. This study comprehensively investigated the quality changes of kelp gel edible granules stored at 4 °C and 25 °C by evaluating indicators such as total bacterial count, coliform bacteria, pH, relaxation time, color difference, appearance, texture characteristics, gel strength, and sensory scoring. The results showed that during the storage at 4 °C, the total bacterial count remained within the national standard range, the hardness and chewiness increased, the gel strength first increased and then decreased, the partial exudation of the bound water in the product occurred, and the sensory score slightly decreased, with an overall minor change in quality. During the storage at 25 °C, significant quality changes were observed, with the total bacterial count exceeding the national standard on the 20th day; additionally, the hardness, chewiness, and gel strength all initially increased and then decreased, both the bound water and the restrained water in the product exuded, the moisture stability decreased, and the sensory score significantly decreased between 16 to 20 days. The spoilage of the product was characterized by a significant water loss, reduction in volume, color change from bright green to dark yellow-brown, and a distinct smell of decaying algae. No coliform bacteria was detected in all products during the storage period. In summary, the shelf life endpoint of the product stored at 25 °C is 16 days, and the shelf life of the product stored at 4 °C is greater than 20 days. Storage at 4 °C can better maintain product quality, extend the shelf life, and effectively maintain the overall color of the product.

## 1. Introduction

Kelp (*Laminaria japonica*) belongs to the brown algae and is one of the important economic algae in large-scale mariculture in China. It is rich in polysaccharides, proteins, vitamins, minerals, iodine, iron, zinc, and more than 60 other nutrients [1], and has various physiological activities such as anti-inflammatory, anti-obesity, anti-thrombotic, and immunomodulatory effects [2,3,4,5]. Kelp has a delicious taste and has a long history of consumption in China, being well-loved by consumers. However, the current kelp processing products in China are mainly primary processed products such as salted, dried, and seasoned kelp strips, with relatively low economic value added. To enhance the economic benefits of kelp processing products, it is necessary to develop new and innovative processed food products. In recent years, the emerging beverage category of milk tea has developed rapidly in the food and beverage industry and has subsequently driven the development of the milk tea ingredient industry. Common milk tea ingredients include pearls, coconut gels, crispy boo boo, and popping bobas. The crispy boo boo is made of konjac gum, and the outer skin of the popping bobas is made of calcium alginate gel. Therefore, the polysaccharide with gel characteristics has the potential to be developed into beverage ingredients. Kelp contains a higher amount of alginate [6], which has unique gel properties. Utilizing this characteristic, a new type of kelp gel edible granules can be developed. These granules have a uniform light green color, an elastic texture, good taste, and a unique flavor of kelp. They are also rich in natural dietary fiber polysaccharides, such as alginate, which are beneficial to intestinal health and can be further processed into ready-to-eat products or added to beverages as ingredients.

The product needs to be circulated, and during this process, issues related to the deterioration of product quality arise. At the same time, temperature plays a crucial role. Appropriate temperature can extend the shelf life of the product, while unsuitable storage temperature can accelerate the spoilage and deterioration of the product [7]. Therefore, studying the quality changes of kelp gel edible granules under different storage temperatures is of great significance for their future practical production. In addition, relying solely on a single evaluation index to reflect the quality change pattern during the storage process is too one-sided and has a low prediction accuracy. It is necessary to measure multiple quality change indicators for comprehensive evaluation. In summary, clarifying the quality characteristics and change patterns of kelp gel edible granules under different storage conditions is the foundation for predicting their shelf life and improving product safety.

This study comprehensively investigated the quality changes of kelp gel edible granules stored at 4 °C and 25 °C by evaluating indicators such as total bacterial count, coliform bacteria, pH, relaxation time, color difference, appearance, texture characteristics, gel strength, and sensory scoring. The aim is to provide basic data for the subsequent industrial application of the product and to provide a theoretical reference for similar products.

## 2. Materials and Methods

### 2.1. Materials

Salted kelp was purchased from Fujian Tianyuan Fisheries Group Co., Ltd. (Fuzhou, Fujian, China); sodium bicarbonate was of food grade and was purchased from Tianjin Bohua Yongli Chemical Co., Ltd. (Tianjin, China); calcium chloride was of food grade and was purchased from Jiangsu Kelunduo Food Ingredients Co., Ltd. (Lianyungang, Jiangsu, China); citric acid was of food grade and was purchased from Huaifang Yingxuan Industrial Co., Ltd. (Huaifang, Shangdong, China); silver nitrate was purchased from Shenzhen Bolinda Technology Co., Ltd. (Shenzhen, Guangdong, China); sodium hydroxide was purchased from Sinopharm Chemical Reagent Co., Ltd. (Shanghai, China); physiological saline was purchased from Fuzhou Haiwang Fuyao Pharmaceutical Co., Ltd. (Fuzhou, Fujian, China); 0.85% sterile physiological saline tubes were purchased from Qingdao High-tech Industrial Park Haibo Biotechnology Co., Ltd. (Qingdao, Shangdong, China); total bacterial count plates and coliform bacteria test plates were purchased from Guangdong Dayuan Oasis Food Safety Technology Co., Ltd. (Guangzhou, Guangdong, China); and pH precision test paper was purchased from Shanghai San Ai Si Reagent Co., Ltd. (Shanghai, China).

### 2.2. Preparation of Kelp Gel Edible Granules

The preparation process of kelp gel edible granules developed by the laboratory in the early stage is as follows [8,9]:

Cleaning: The salted kelp was cleaned using a sponge brush in tap water, then rinsed with pure water.

Desalting: The salted kelp was desalted by soaking. Cleaned salted kelp (150 g) was soaked in 4 L of pure water at 25 °C for 12 h. The pure water was changed every 3 h (stirred once every hour). AgNO_3_ solution (0.1 mol/L) was added to an appropriate amount of kelp soaking liquid to observe if there is a white precipitate, and a small piece of desalted kelp was tasted. The desalting process was completed when no significant precipitate formed with the addition of 0.1 mol/L AgNO_3_ solution, and the desalted kelp was tasted without obvious saltiness. 

Deodorizing: The desalted kelp was cut into evenly sized strips and deodorized for 90 min at 25 °C with 3% β-cyclodextrin, then rinsed three times with pure water to remove excess deodorizing liquid.

Mincing: The deodorized kelp was minced into small pieces using a meat grinder (TK-12, Jinhua, Zhejiang, China). 

Heating and converting: A total of 3% of the minced kelp’s weight of sodium bicarbonate and 100 mL of pure water were added into minced kelp (100 g). The mixture was placed in a constant temperature water bath pot (HWS-28, Shanghai, China) at 100 °C and stirred continuously for 11 min.

Fine grinding: Pure water at a ratio of 1: 1.1 (kelp to water, m:v) was added into the heated kelp mixture and grinded for 5 min using a wall breaker (L18-Y68S, Jinan, Shangdong, China).

Solidification and molding: An appropriate amount of kelp paste was placed into a round mold with a pipette and demolished in 2% calcium chloride solution at 20 °C and pH 8 and stood for 4 min to set.

Rinsing: After solidification, the kelp gel edible granules were quickly removed and placed in pure water to wash off excess calcium chloride.

Vacuum packaging: The rinsed kelp gel edible granules and 30 mL of pure water were vacuum sealed in a pure aluminum foil high-temperature steaming bag for subsequent experiments.

Heat treatment: The kelp gel edible granules of equal weight packaged in aluminum foil cooking bags were sterilized at 90 °C for 5 min.

Storage conditions: The kelp gel edible granules were stored at 4 °C and 25 °C, respectively. The total bacterial count, coliform bacteria, pH, relaxation time, color difference, appearance, size, texture characteristics, gel strength, and sensory scoring of the products were measured every four days.

### 2.3. Determination of Sterilization Parameters

The products of equal weight packaged in aluminum foil cooking bags were sterilized using different sterilization parameters (80, 85, 90, and 95 °C for 5 min and 10 min) in hot water in a constant temperature water bath pot (HWS-28, Shanghai, China). After sterilization, the products were immediately cooled in running water for 30 min. Finally, the products were placed in an incubator at 37 °C for 7 days, after which the bacterial count, coliform bacteria, hardness, and appearance were measured.

### 2.4. Determination of Total Bacterial Count

The total bacterial count was determined according to GB 4789.2-2022 [10]. Minced kelp gel edible granules (5 g) were placed into a sterile homogenization bag containing 45 mL of physiological saline. The mixture was homogenized and evenly patted for 90 s to create a 1:10 (g:mL) sample homogenate. A total of 1 mL of the homogenate was added to a tube containing 9 mL of 0.85% physiological saline to create a 1:100 sample homogenate. This procedure was repeated to prepare 2–3 additional appropriate dilution samples. A total of 1 mL of the appropriately diluted sample homogenate was dropped vertically and uniformly onto the center of the total bacterial count test plate. The test plates were incubated at 37 °C for 24 h. After incubation, the total bacterial count was calculated. 

### 2.5. Determination of Coliform Bacteria

The coliform bacteria was determined according to the second method outlined in GB 4789.3-2016 [11]. The sample homogenate was prepared using the same method as described in the total bacterial count determination. The pH of the sample homogenate was adjusted to between 6.5 and 7.5 using 1 mol/L NaOH. A total of 1 mL of the appropriately diluted sample homogenate was dropped vertically and uniformly onto the center of the coliform bacteria test plate. The test plates were incubated at 37 °C for 24 h.

### 2.6. Determination of pH

The pH measurement was conducted with slight modifications based on GB 5009.237-2016 [12]. Minced kelp gel edible granules (1 g) were mixed with 9 mL of distilled water. The mixture was shaken for 30 s and then extracted for 30 min. The mixture was filtered to obtain the filtrate. The pH of filtrate was measured using a pH meter (Five Easy Plus FE28, Mettler Toledoand, Zurich, Switzerland) and the value was recorded.

### 2.7. Low-Field Nuclear Magnetic Resonance Analysis

LF-NMR relaxation tests were performed according to the method of Li, with some modifications [13]. A kelp gel edible granule (about 0.65 g) was dried to remove surface moisture and placed in a 40 mm NMR tube (NM120-069H-1, Suzhou, Jiangsu, China). The *T*_2_ relaxation time was measured using the Carr–Purcell–Meiboom–Gill (CPMG) sequence. The echo time, wait time, and the number of scans were set to 0.5 ms, 5000 ms, and 4, respectively. A total of 4000 echoes were acquired for analysis. The obtained data were inverted using Multi Exp Inv Analysis software Version 4.0.

### 2.8. Color Difference Determination

The color of individual kelp gel edible granule was measured using a colorimeter (CR9, Guanzhou, Shenzhen, China) in Specular Component Include (SCI) mode and the color values were represented with L* (lightness), a* (red-green value; positive for red, negative for green), and b* (yellow-blue value; positive for yellow, negative for blue). The total color difference (ΔE) was evaluated using Equation (1).
(1)$$∆E=\sqrt{{\left(L-L_{0}\right)}^{2}+{\left(a-a_{0}\right)}^{2}+{\left(b-b_{0}\right)}^{2}}$$
where L_0_, a_0_, and b_0_ are the values on the 0th day of storage.

### 2.9. Texture Profile Analysis (TPA)

Texture profile analysis tests were performed according to Cui with some modifications [14]. The TPA tests were performed using a texture analyzer (TA-XT plus, Stable Micro System, Godalming, UK) with a cylindrical steel P/36R probe at 30% compression. The pre-test speed, test speed, and post-test speed were all set to 1.0 mm/s. The trigger force was set to 5 g. The test was performed twice with a 5 s interval between measurements. The gel strength was evaluated using Equation (2) according to Liu [15].
(2)$$Gel\;Strength(g/{cm}^{2})=\frac{Hardness}{Contact\;Area}$$

### 2.10. Sensory Quality Determination

The sensory quality of the kelp gel edible granules was evaluated using a quantitative descriptive analysis (QDA) method. An evaluation panel consisting of eight food science graduate students (4 males and 4 females) was organized to taste the kelp gel edible granules. The evaluation was based on three dimensions: appearance, texture, and flavor, with weights set at 30%, 40%, and 30%, respectively. The comprehensive score was calculated by removing the highest and lowest scores and then averaging the remaining scores (Table 1). 

### 2.11. Data Processing

Data were organized using EXCEL software Version 2021. Single-factor ANOVA tests were performed using SPSS 27.0.1 software. Graphs were created using Origin 2023b software. Each measurement was conducted with at least three replicates, and results are presented as mean ± SEM. A correlation analysis was performed using Pearson’s test to determine the direct and indirect contributions of these indicators to the quality deterioration of products during storage.

## 3. Results

### 3.1. Determination of Sterilization Process Parameters

The products were sterilized under different parameters and stored at 37 °C for 7 days. It was observed that at a sterilization temperature of 80 °C, both 5 min and 10 min treatments resulted in significant quality deterioration (Table 2). The products showed a deepened color, volume shrinkage, and increased hardness. At a sterilization temperature of 85 °C, the products maintained a normal appearance, but hardness still increased. When the sterilization temperature was above 85 °C, the products retained their light green color, volume remained unchanged, hardness did not significantly increase, and no total bacterial count was detected. With the increase in sterilization temperature and sterilization time, the less the total bacterial count, the better the quality of products. Therefore, a sterilization temperature of 90 °C for 5 min was selected as the parameter for subsequent product storage tests.

### 3.2. Changes in Total Bacterial Count and Coliform Bacteria during Storage

The growth rate of the total bacterial count in products stored at 4 °C was slower compared to those stored at 25 °C. By the end of the storage period, the total bacterial count in the 4 °C stored products did not exceed the limit of 4.48 log10 (CFU/g) as per GB 19643-2016 [16]. For products stored at 25 °C, the total bacterial count exhibited two distinct phases. In the first phase, microbial growth was relatively slow, while in the second phase, the growth rate accelerated. From 4 to 8 days, the bacterial count increased slowly, but from 8 to 20 days, the growth rate rapidly accelerated, reaching 4.62 log10 (CFU/g) on the 20th day, surpassing the GB 19643-2016 limit [16]. At this point, the product showed signs of spoilage, including severe moisture loss, reduced volume, color change from bright green to dark yellow-brown, and a foul algae-like odor (Figure 1). Throughout the entire storage period, the coliform bacteria was not detected, indicating that the product was not contaminated by fecal matter during production and maintained good sanitary conditions

### 3.3. Changes in pH during Storage

The initial pH of all groups was around 9.35. The pH value of the products stored at 4 °C showed minimal fluctuation throughout the storage period. In contrast, the pH of products stored at 25 °Cexhibited two distinct phases: a slight decrease from 0 to 8 days and a significant drop from 8 to 20 days, ultimately reaching 6.96 on the 20th day (Figure 2).

### 3.4. Changes in Moisture Distribution during Storage

*T*_21_ represents the chemically bound water in the product (bound water), and *T*_22_ represents the water bound within the gel network (restrained water) of the product (accounting for most of the product), which is more unstable compared to *T*_21_ and can transform into free water and be lost. P_21_ and P_22_ indicate the proportion of water with relaxation times *T*_21_ and *T*_22_ in the total moisture of the product, respectively. It could be seen that *T*_21_ of the product stored at 4 °C did not change significantly throughout the storage period, consistent with the findings of Huang [17] (Figure 3a). However, *T*_22_ shortened from day 4 to day 20 compared to day 0, indicating that restrained water within the gel network in the 4 °C product became more stable during storage. P_21_ in the product stored at 4 °C tended to exude, forming P_21-1_, which was less stable than P_21_ but more stable than P_22_ (Figure 3b). This suggested that the stability of bound water decreased while the stability of restrained water within the gel network increased during storage at 4 °C. It could be seen that *T*_21_ of the product stored at 25 °C gradually shifted to the right, while *T*_22_ first shifted to the left and then to the right (Figure 4a). The number of water types in the product stored at 25 °C changed from two to four. During the storage period from day 4 to day 16, *T*_21_ exuded, forming the P_21-1_ component. In the later storage period (day 16 to day 20), the stability of P_22_ decreased, and it exuded as P_22-1_, whose relaxation time significantly increased, indicating a rapid decrease in stability and a tendency to be lost (Figure 4b).

### 3.5. Changes in Color Difference during Storage

ΔE represents the overall color change, with larger ΔE values indicating more severe deterioration. The ΔE values for products stored at both 4 °C and 25 °C increased over time, indicating a greater overall color change during storage. On the 20th day, the total color difference of products stored at 25 °C was nearly twice than products stored at 4 °C. The decrease in L* and a* values in products stored at 4 °C contributed to the increase in ΔE, while b* remained unchanged. For products stored at 25 °C, L*, a*, and b* all changed significantly, reflecting a shift in appearance from bright green to dark yellow-brown (Table 3). This demonstrated that low-temperature storage better preserved the overall color of the product.

### 3.6. Changes in Appearance during Storage

The color of the products stored at 4 °C changed from the initial green to dark green, with reduced brightness, consistent with the changes in a* and ΔE values. For products stored at 25 °C, the color initially shifted from green to dark green and eventually turned dark brown by the 16th day, aligning with the changes in a*, b*, and ΔE values. The significant color change in appearance was related to the significant increase in total bacterial count and the decrease in pH discussed earlier (Figure 5).

By the 20th day of storage, the volume of spoiled products significantly decreased compared to normal products. This observation aligned with the changes in relaxation time and moisture ratio. During spoilage, the *T*_22_ relaxation time gradually increased, indicating the decreased stability of restrained water within the gel network. As restrained water, which constitutes the majority of the moisture of products, leaked out and transformed into less stable free water, a reduction in restrained water in the gel network and consequently a decrease in product volume occurred (Figure 6).

### 3.7. Changes in Texture and Gel Strength during Storage

The hardness of products stored at 4 °C significantly increased from 0 to 4 days and remained relatively stable from 4 to 20 days, with an overall increase of nearly 50% compared to day 0. Chewiness, elasticity, resilience, and gel strength generally showed a trend of first increasing and then decreasing, while cohesiveness showed no significant difference. For products stored at 25 °C, the changes in hardness and chewiness exhibited three phases: First, from 0 to 4 days, hardness and chewiness rapidly increased, with both being 1.5 times higher on day 4 compared to day 0. Second, from 4 to 12 days, hardness slightly decreased, and chewiness remained relatively unchanged. Third, from 12 to 20 days, the products underwent spoilage, resulting in a rapid decrease in hardness and chewiness, while elasticity and cohesiveness increased, and resilience showed no significant difference. Gel strength increased gradually from 0 to 8 days and then decreased after 8 days (Table 4).

### 3.8. Changes in Sensory Quality during Storage

The color, odor, texture, and comprehensive sensory scores of the product stored at 4 °C changed little, and the comprehensive sensory score remained between 8.2 and 7.5; throughout the entire storage process, the kelp flavor of the 4 °C product was lighter, but the texture was tight, and the color remained green. Although ΔE indicated that the overall color of the product had changed, the change was not significant enough for the average observer to discern (Figure 7a).

The color score of the 25 °C product decreased from 8.17 to 2.97; the odor score decreased from 7.48 to 4.23; the texture score decreased from 8.37 to 3.75; and the comprehensive sensory score decreased from 8.04 to 3.66 (Figure 7b).

### 3.9. Correlation Analysis of Quality Indicators of Kelp Gel Edible Granules

The total bacterial count and ΔE showed a highly significant positive correlation with storage time and the comprehensive sensory score showed a significant negative correlation with the storage time of the 4 °C stored product. For the 25 °C stored product, the total bacterial count and ΔE had a highly significant positive correlation with storage time; pH and overall sensory score had a highly significant negative correlation with storage time.

The above results indicated that the total bacterial count, ΔE, and comprehensive sensory score of the 4 °C stored product all changed significantly with the change in storage time. In addition to the above indicators, the pH of the 25 °C stored product could also reflect the change in product quality with storage time. Moreover, the key quality indicator of the product stored at different temperatures was different except for the comprehensive sensory evaluation. The key indicator for the 4 °C stored product was the total bacterial count (correlation coefficient was 0.977), and the key indicator for the 25 °C stored product was pH (correlation coefficient was −0.965) (Table 5).

## 4. Discussion

### 4.1. Microbiological Analysis

The total bacterial count is an important indicator for assessing the hygienic condition of processed products, while coliform bacteria is commonly used as an indicator of fecal contamination. Both indicators reflect the sanitary status of food and are crucial for determining its edibility. Kelp can be contaminated with human pathogens during cultivation, such as *Vibrio* spp., *Salmonella* spp., *Staphylococcus aureus*, *Listeria*, etc. Additionally, the introduction of pathogens uncommon to the marine environment, such as *L. monocytogenes*, during handling and post-harvesting processing into finished products can be another route for kelp contamination [1,18]. The observed changes in the total bacterial count at 25 °C might be due to the initial low bacterial count after pasteurization, which killed most microorganisms mentioned above except for some heat-resistant spores. These spores might be some of the thermostable Bacillus species, such as *Bacillus pumilus* and *B. licheniformis* [19]. As storage time progressed, these spores began to germinate but adapted slowly to the environment, leading to a slower initial growth rate. In the later stages, microorganisms adapted to the environment and rapidly proliferated using the abundant materials such as alginate polysaccharides present in the product. In summary, low temperatures effectively inhibit the metabolic activity and proliferation of microorganisms. Similarly, several studies have reported that microorganisms grew more slowly at lower temperatures [20,21].

### 4.2. pH Analysis

The changes in products stored at 25 °C could be explained by the initial heat sterilization stage at 25 °C, which killed most microorganisms, resulting in a low acid production and minimal pH change in the early stage. After 8 days, the microbial growth rate increased, the total bacterial count rose, and the microorganisms utilized organic substances such as alginate polysaccharides, producing organic acids that led to a decrease in pH. It was noteworthy that the pH changes did not exhibit the typical “V” shape (initial decrease followed by an increase) often seen in food products [22]. This was primarily due to the low protein content in the kelp gel particles. During spoilage, the decomposition of proteins into alkaline substances like ammonia and amines was insufficient to neutralize the acidic substances produced from the decomposition of polysaccharides and other organic materials. Consequently, the pH value of the product continued to decline throughout the storage period.

### 4.3. Moisture Distribution Analysis

The observed phenomena of moisture distribution could be attributed to several factors. First, for products stored at 4 °C, the significant increase in hardness during storage might cause the three-dimensional gel network to become finer and more uniform, increasing the capillary binding force of the gel network and thus increasing the stability of restrained water, resulting in the leftward shift of *T*_22_. Second, for products stored at 25 °C, the initial leftward shift of *T*_22_ could be attributed to the same reasons mentioned above. After 16 days of storage, *T*_22_ began to shift to the right. At this point, the internal microbial proliferation rate increased, and bacteria decomposed alginate, leading to the destruction and disintegration of some parts of the gel network. The damaged gel network reduced its water-binding capacity [23], causing some restrained water to exude from the disintegrated network as free water. Additionally, kelp gel edible granules are hydrogels with a three-dimensional network of an “egg box” model formed by cross-linking COO- and Ca^2+^ in sodium alginate [24]. A large amount of H^+^ produced by the bacterial decomposition of alginate and other organic matter might compete with Ca^2+^ (H^+^ and COO- will protonate to form carboxylic acid) [25], thereby replacing Ca^2+^ from calcium alginate to produce alginate acid, which will also lead to the disintegration of the gel network. This ultimately resulted in the leftward shift of *T*_22_ and the formation of P_22-1_ water in the later storage period.

### 4.4. Physicochemical Properties

Color can reflect the sensory and physicochemical changes in a product and is an important indicator of product acceptability [26]. This observed phenomenon of color could be attributed to the structure of chlorophyll, which consists of a planar ring formed by one porphyrin ring and four pyrrole subunits. The electrons can migrate freely within this structure, making chlorophyll unstable under conditions such as acid, heat, light, and oxygen [27]. Research has shown that pheophytinisation was associated with the number of acids liberated during processing. Once an acidic environment (pH < 7) surrounded the chlorophyll compounds, the structural change in the chlorophyll molecule affected the overall structure of the macroalgae, which led to a change in the color of the macroalgae [28]. During storage at 25 °C, the products underwent spoilage, and microorganisms decomposed organic matter to produce acids, leading to the replacement of Mg^2+^ in chlorophyll with H^+^, forming yellow-brown pheophytin [29]. Conversely, the products stored at 4 °C had slower microbial growth, maintaining a pH above 9, and thus the green retention was good. In summary, low-temperature storage can inhibit microbial growth and reduce the damage caused by acid and heat to chlorophyll in the product, thereby better preserving the overall color.

The reasons for the changes in texture characteristics might be as follows: First, the kelp paste contained dispersed sodium alginate, which, during the demolding and curing process, was attracted by the Ca^2+^ in calcium chloride to the surface of the product for a replacement reaction, resulting in an uneven distribution of sodium alginate inside the product (more on the surface) [30]. As the storage time extended, Ca^2+^ gradually diffused from the surface to the inside of the product, and sodium alginate also gradually distributed evenly in this process. The three-dimensional network gel structure inside the product became more uniform and dense, thereby enhancing the overall deformation resistance, which ultimately led to a significant increase in all product hardness and chewiness in the first 0–4 days [25,31]. Since a stable gel structure had been formed inside the product during this period, the hardness and chewiness of the product stored at 4 °C did not change much from 4 to 20 days. Second, the quality of the 25 °C product deteriorated significantly in the later stage of storage, and the pH decreased significantly during the massive proliferation of microorganisms; the produced H^+^ might replace some Ca^2+^ and cause the alginate calcium gel to transform into alginate gel. Since ionic bonds have higher bond energy than hydrogen bonds [32], the mechanical properties of the alginate calcium gel were stronger than alginate gel, and the gel network structure collapsed during the spoilage process of the product, which comprehensively led to a significant decrease in hardness and chewiness. As the hardness decreased, elasticity and cohesiveness increased. Finally, since the gel strength is a composite index based on hardness and product area, its changes during the storage process were consistent with the reasons for the changes in hardness. A previous study has also reported that table grapes could better maintain their hardness under low temperature conditions [33].

### 4.5. Sensory Quality Analysis

Generally, a mean liking score of ≥7 on a 9-point hedonic scale is associated with a highly acceptable sensory quality [34]. When the products were stored at 4 °C, the color, odor, texture, and comprehensive sensory scores were kept between 8.2 and 7.5, indicating that the products were highly acceptable. While the products were stored at 25 °C, the color, odor, texture, and comprehensive sensory scores were all lower than 7 after 8 days of storage. The main reason for the significant decrease in the sensory score of the 25 °C product in the later stage of storage was the massive proliferation of microorganisms. This was consistent with the above reasons for changes in color, texture, and total bacterial count. A previous study has also reported that under low-temperature conditions, the pH change of beef meatballs was slower, and the starting point of sensory quality change was longer [35].

### 4.6. Correlation Analysis of Quality Indicators Analysis

The purpose of correlation analysis is to determine the direct and indirect contribution of these quality indicators to the quality deterioration of products in the storage process through the correlation coefficient, and to provide an indicator basis for monitoring product quality changes in the future. The key indicator of products stored at 4 °C was the total bacterial count, which was significantly positively correlated with storage time, indicating that low temperature only slowed down the growth rate of microorganisms. The key indicators of the products stored at 25 °C were pH and total bacterial count, which were significantly positively correlated with storage time. The change in pH was caused by the change in total bacterial count.

## 5. Conclusions

This study comprehensively investigated the quality changes of kelp gel edible granules stored at 4 °C and 25 °C by evaluating indicators such as total bacterial count, coliform bacteria, pH, relaxation time, color difference, appearance, texture characteristics, gel strength, and sensory scoring. Data gathered from this research show that the shelf life endpoint for products stored at 25 °C is 16 days, while for products stored at 4 °C, it is more than 20 days. Low-temperature storage (4 °C) better maintains the quality of kelp gel edible particles, extends their shelf life, and effectively preserves the overall color of the product. Future studies are warranted to flavor kelp gel edible particles to enrich their flavor substances, develop beverages adding kelp gel edible particles, enrich their application forms, and expand the application range in the market.

## Figures and Tables

**Figure 1 foods-13-02267-f001:**
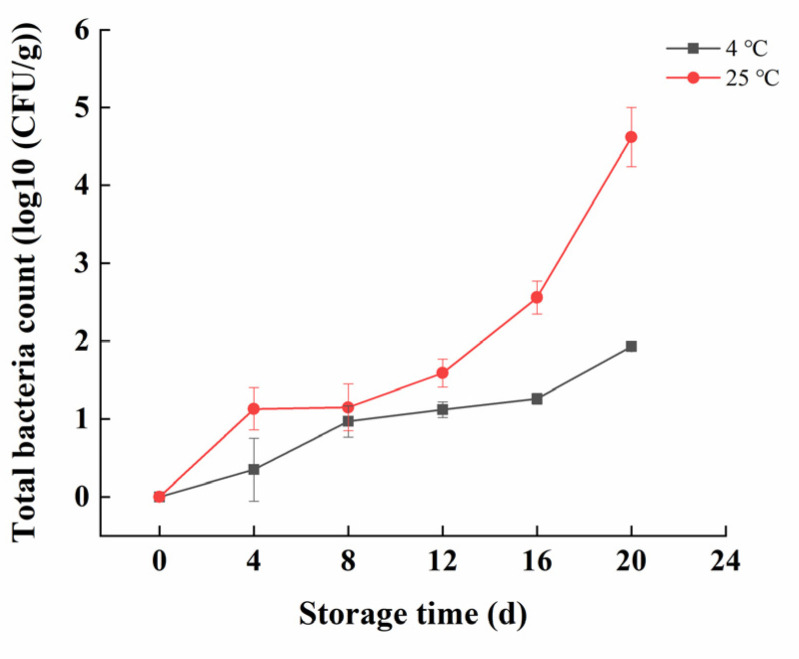
Changes in the total bacteria count of kelp gel edible granules during storage times.

**Figure 2 foods-13-02267-f002:**
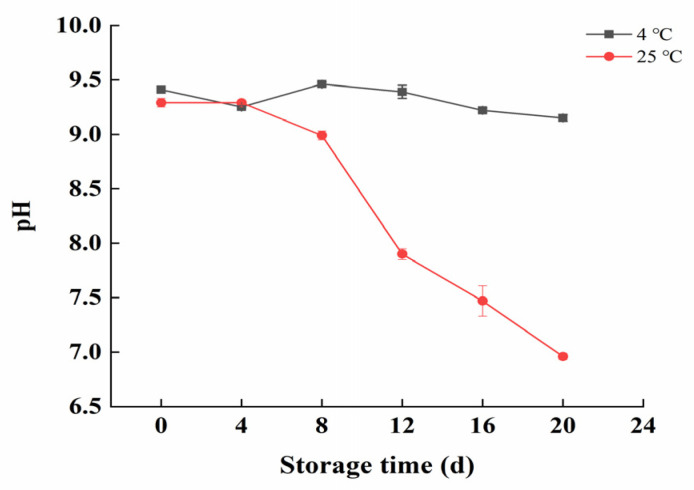
Changes in the pH of kelp gel edible granules during storage times.

**Figure 3 foods-13-02267-f003:**
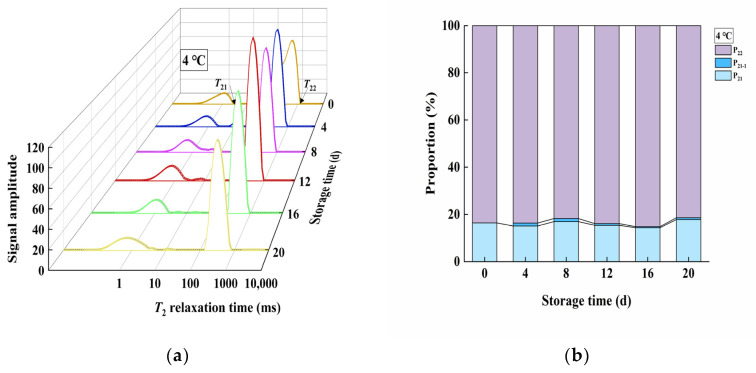
Changes in the water distribution of kelp gel edible granules under 4 °C during storage times. (**a**) Change in relaxation time of the product during storage at 4 °C; (**b**) change in water distribution ratio during storage at 4 °C.

**Figure 4 foods-13-02267-f004:**
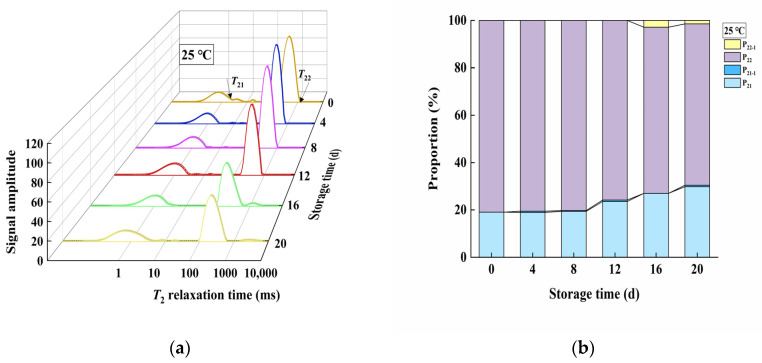
Changes in the water distribution of kelp gel edible granules under 25 °C during storage. times. (**a**) Change in relaxation time of the product during storage at 25 °C; (**b**) change in water distribution ratio during storage at 25 °C.

**Figure 5 foods-13-02267-f005:**
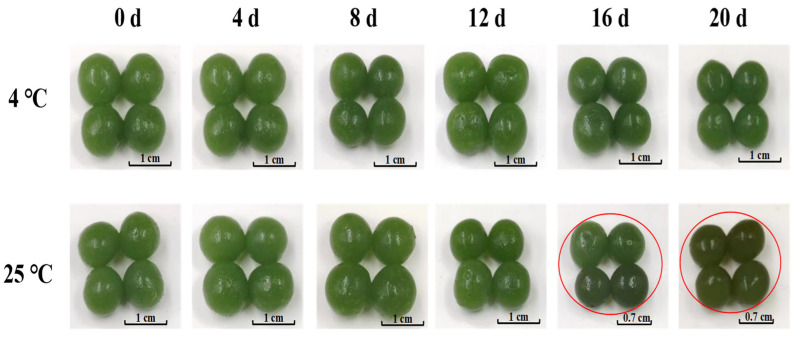
Changes in the appearance of kelp gel edible granules during storage time. The pictures circled in red indicated that the appearance of the products had changed significantly.

**Figure 6 foods-13-02267-f006:**
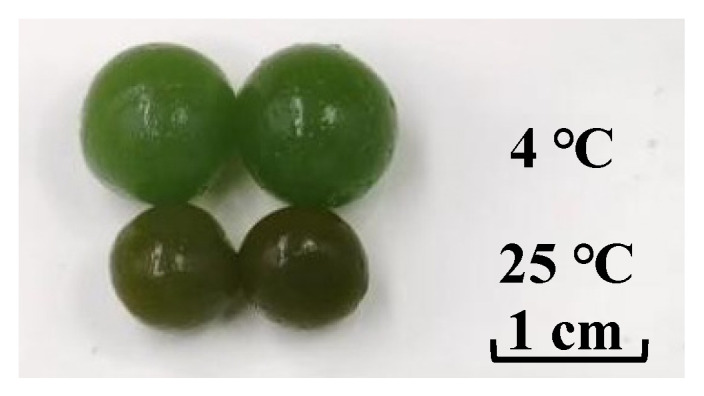
Appearance comparison of kelp gel edible granules on the 20th day of storage.

**Figure 7 foods-13-02267-f007:**
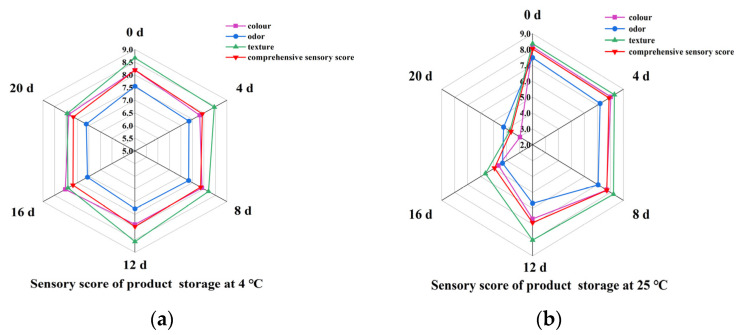
Changes in sensory quality of kelp gel edible granules during storage times. (**a**) Change in quality of the product during storage at 4 °C; (**b**) change in sensory quality of the product during storage at 25 °C.

**Table 1 foods-13-02267-t001:** Sensory standard of kelp gel edible granules during storage.

Attribute	Scoring Criteria	Score Range
Color (30%)	Green	8.1~10
	Slightly faded green or deeper green	6.1~8
	Light brown or dark green	4.1~6
	Brown	2.1~4
	Dark brown	0~2
Odor (40%)	Strong kelp aroma, no off-flavor	8.1~10
	Mild kelp aroma, no off-flavor	6.1~8
	No kelp aroma, no off-flavor	4.1~6
	Slight off-flavor	2.1~4
	Strong off-flavor	0~2
Texture (30%)	Good elasticity, firm texture	8.1~10
	Fair elasticity, firm texture	6.1~8
	Average elasticity, relatively firm texture	4.1~6
	Poor elasticity, soft or hard texture	2.1~4
	No elasticity, very soft or very hard texture	0~2

**Table 2 foods-13-02267-t002:** Effect of different sterilization parameters on products quality.

Sterilization Conditions		Microbial Indicators		Quality Changes	
Sterilization Temperature (°C)	Sterilization Time (min)	Total Bacterial Count (CFU/g)	Coliform Bacteria (CFU/g)	Hardness (g)	Appearance
80	5	142.50 ± 41.13 ^a^	-	162.68 ± 5.25 ^a^	Darkened color, volume shrinkage
	10	115.00 ± 17.32 ^ab^	-	155.40 ± 1.16 ^b^	
85	5	67.50 ± 35.94 ^bc^	-	140.74 ± 8.17 ^c^	Darkened color, volume shrinkage
	10	60.00 ± 40.99 ^c^	-	133.97 ± 2.61 ^c^	Light green, no volume change
90	5	-	-	116.99 ± 1.32 ^d^	Light green, no volume change
	10	-	-	110.32 ± 1.28 ^d^	Light green, no volume change
95	5	-	-	115.38 ± 4.40 ^d^	Light green, no volume change
	10	-	-	115.74 ± 2.50 ^d^	Light green, no volume change

“-” indicates not detected. Each value is the mean ± standard deviation (*n* = 8). Superscripts: different small letters within rows indicate significant difference among treatments (*p ≤* 0.05).

**Table 3 foods-13-02267-t003:** Color parameters and changes of kelp gel edible granules during storage times.

Storage Temperature	Storage Days(d)	L*	a*	b*	ΔE
4 °C	0	27.09 ± 1.46 ^a^	−11.36 ± 1.03 ^ab^	12.57 ± 2.39 ^a^	
	4	22.67 ± 0.46 ^b^	−13.28 ± 1.03 ^bc^	14.63 ± 1.41 ^a^	5.32 ± 1.30 ^c^
	8	22.91 ± 0.35 ^b^	−13.07 ± 0.84 ^bc^	13.26 ± 2.16 ^a^	4.95 ± 0.32 ^c^
	12	21.73 ± 0.57 ^b^	−14.23 ± 0.11 ^c^	14.80 ± 1.26 ^a^	6.55 ± 0.08 ^bc^
	16	19.74 ± 0.92 ^c^	−13.57 ± 0.75 ^c^	11.65 ± 1.33 ^a^	7.82 ± 0.98 ^ab^
	20	23.21 ± 0.12 ^b^	−10.94 ± 1.34 ^a^	9.65 ± 2.62 ^a^	8.44 ± 0.88 ^a^
25 °C	0	31.20 ± 1.89 ^a^	−10.95 ± 0.67 ^b^	13.14 ± 0.68 ^ab^	
	4	25.38 ± 1.42 ^b^	−13.48 ± 0.31 ^c^	16.27 ± 2.55 ^a^	7.47 ± 0.34 ^d^
	8	22.47 ± 2.40 ^bc^	−13.90 ± 0.94 ^c^	15.18 ± 4.59 ^a^	10.37 ± 0.17 ^c^
	12	20.50 ± 1.00 ^cd^	−13.51 ± 0.72 ^c^	11.69 ± 3.96 ^abc^	11.58 ± 0.94 ^c^
	16	17.17 ± 0.47 ^de^	−11.79 ± 1.72 ^b^	8.93 ± 0.52 ^bc^	14.73 ± 0.20 ^b^
	20	15.60 ± 2.36 ^e^	−8.93 ± 0.40 ^a^	6.60 ± 0.55 ^c^	17.08 ± 1.94 ^a^

Each value is the mean ± standard deviation (*n* = 6). Superscripts: different small letters within rows indicate significant difference among treatments (*p ≤* 0.05).

**Table 4 foods-13-02267-t004:** Changes in TPA and gel strength of kelp gel edible granules during storage times.

Storage Temperature	Storage Days (d)	Hardness (g)	Elasticity	Cohesiveness	Chewiness (g)	Recoverability	Gel Strength (g/cm^2^)
4 °C	0	107.31 ± 3.62 ^c^	0.74 ± 0.03 ^c^	0.72 ± 0.02 ^a^	57.00 ± 4.09 ^c^	0.46 ± 0.01 ^a^	96.8 ± 10.29 ^d^
	4	145.30 ± 6.54 ^ab^	0.81 ± 0.01 ^a^	0.73 ± 0.02 ^a^	85.89 ± 4.92 ^ab^	0.47 ± 0.01 ^a^	135.21 ± 6.08 ^b^
	8	151.54 ± 8.27 ^a^	0.81 ± 0.01 ^a^	0.73 ± 0.01 ^a^	89.05 ± 5.33 ^a^	0.47 ± 0.01 ^a^	128.17 ± 12.65 ^bc^
	12	141.47 ± 2.00 ^b^	0.81 ± 0.01 ^a^	0.72 ± 0.01 ^a^	82.34 ± 2.38 ^b^	0.46 ± 0.01 ^a^	157.30 ± 10.15 ^a^
	16	146.33 ± 5.05 ^ab^	0.79 ± 0.01 ^ab^	0.71 ± 0.01 ^a^	81.48 ± 5.28 ^b^	0.45 ± 0.01 ^ab^	128.81 ± 9.67 b ^c^
	20	151.42 ± 3.03 ^a^	0.78 ± 0.02 ^b^	0.70 ± 0.02 ^a^	83.19 ± 3.38 ^b^	0.44 ± 0.01 ^b^	117.90 ± 5.52 ^c^
25 °C	0	96.38 ± 3.06 ^c^	0.75 ± 0.02 ^d^	0.71 ± 0.01 ^c^	50.69 ± 1.39 ^d^	0.46 ± 0.00 ^a^	89.51 ± 4.03 ^c^
	4	144.33 ± 4.31 ^a^	0.80 ± 0.02 ^c^	0.73 ± 0.02 ^c^	84.47 ± 5.54 ^b^	0.46 ± 0.02 ^a^	134.31 ± 4.01 ^a^
	8	150.59 ± 4.78 ^a^	0.81 ± 0.01 ^c^	0.72 ± 0.01 ^c^	87.30 ± 4.28 ^ab^	0.45 ± 0.01 ^a^	135.13 ± 11.32 ^a^
	12	135.86 ± 8.20 ^b^	0.84 ± 0.02 ^ab^	0.80 ± 0.03 ^b^	90.73 ± 4.83 ^a^	0.47 ± 0.02 ^a^	112.92 ± 7.28 ^b^
	16	86.07 ± 3.42 ^d^	0.82 ± 0.01 ^bc^	0.82 ± 0.02 ^b^	58.04 ± 4.59 ^c^	0.45 ± 0.02 ^a^	115.17 ± 11.05 ^b^
	20	86.67 ± 7.41 ^d^	0.86 ± 0.02 ^a^	0.84 ± 0.01 ^a^	62.96 ± 6.15 ^c^	0.46 ± 0.01 ^a^	120.71 ± 10.39 ^b^

Each value is the mean ± standard deviation (*n* = 6). Superscripts: different small letters within rows indicate significant difference among treatments (*p ≤* 0.05).

**Table 5 foods-13-02267-t005:** Correlation analysis results of various indexes of kelp gel edible granules.

Storage Temperature	Measurement Indicator	Storage Time	Total Bacterial Count	ΔE	pH	Hardness	Chewiness	Gel Strength	Comprehensive Sensory Score
4 °C	Storage time	1	0.977 **	0.907 **	−0.632	0.682	0.515	0.310	−0.877 *
	Total bacterial count		1	0.879 *	−0.537	0.725	0.569	0.302	−0.858 *
	ΔE			1	−0.653	0.879 *	0.780 *	0.574	−0.906 **
	pH				1	−0.402	−0.246	0.030	0.690
	Hardness					1	0.972 **	0.597	−0.843 *
	Chewiness						1	0.696	−0.701
	Gel strength							1	−0.236
	Comprehensive Sensory score								1
25 °C	Storage time	1	0.934 **	0.962 **	−0.965 **	−0.423	−0.045	0.243	−0.930 **
	Total bacterial count		1	0.889 **	−0.900 **	−0.484	−0.147	0.267	−0.954 **
	ΔE			1	−0.864 *	−0.200	0.167	0.490	−0.830 *
	pH				1	0.584	0.208	0.006	0.949 **
	Hardness					1	0.912 **	0.600	0.694
	Chewiness						1	0.689	0.375
	Gel strength							1	−0.037
	Comprehensive Sensory score								1

* indicates significant difference at the *p* < 0.05 level; ** indicates extremely significant difference at the *p* < 0.01 level.

## Data Availability

The original contributions presented in the study are included in the article, further inquiries can be directed to the corresponding author.

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
