# Peer review of "Study on Quality Changes of Kelp Gel Edible Granules during Storage"

_foods, 2024, doi:10.3390/foods13142267_

Round 1

Reviewer 1 Report

Comments and Suggestions for Authors

The manuscript (foods-3028972) deals with an interesting product’s (kelp gel granules) quality and stability. Most sections of the manuscript need to be enhanced. Selected overall recommendations are suggested as follows:

Introduction. Improve the section, selected questions can help you. Justify the potential uses and applications of the kelp gel granules. Currently, are there any similar products in the market? Are consumers are interested in this type of product?

L83. Move the number of cites from title to the next line and introduce them briefly.

L93. Change “cutted” to “cut”.

Section 2. Include one subsection to describe the heat treatment and the storage conditions of the kelp gel granules. Also, mention which analysis were performed and their frequency. The size of the kelp gel granules should be analyzed.

Section 2.3. The heat treatment corresponding to a pasteurization process than a sterilization. Please, revise and correct.

L115. In brief, include the heat treatment conditions (temperatures and times), the medium to heat (hot water or vapor) or the equipment to apply the treatment.

L117. List the relevant indicators you measured.

L126. Include the trademark and the manufacturer of the total bacterial count test plate.

L133-134. Provide the trademark and the manufacturer of the total bacterial count test plate.

L150-151. Mention the colorimeter’s mode.

Section 2.11. Describe the type of correlation analysis you performed and its objective.

L218-222. Change “lg” by “log”.

L228. Check and correct the Figure and Figure title format.

L279. Modify “that of” by “than”.

L286. Include “Color parameters and” in the Table’s 3 title.

L323 and L328. The increases are ~50% more, rather than1.5 times. Please, revise and correct.

Figure 7. Please, check the format of Figures and include the missing one.

L380-383. Modify the discussion, considering the heat treatment (pasteurization instead of sterilization). The spores do not revive, probably germinate. Provide information about the typical microorganisms of kelp or the product to explain which spores are present. Include references.

L412-415. At what pH occur the alginate decomposition by acids? Include references and improve the discussion.

L423-425. Provide the pH value of chlorophyll changes.

L434-438. Add adequate references.

Discussion about Sensory quality and Correlation analysis of quality are missing.

Conclusions. Please, do not summary the results, rewrite the conclusions and mention the future directions of the research.

Author Response

Comments 1: Introduction. Improve the section, selected questions can help you. Justify the potential uses and applications of the kelp gel granules. Currently, are there any similar products in the market? Are consumers interested in this type of product?

Response 1: Thank you for pointing this out. We agree with this comment. Therefore, we have added introductions to some similar products in the market and their industry status. This modification has been done as follows:

“In recent years, the emerging beverage category of milk tea has developed rapidly in the food and beverage industry, and has subsequently driven the development of the milk tea ingredient industry. Common milk tea ingredients include pearls, coconut gels, crispy boo boo and popping bobas. The crispy boo boo are made of konjac gum, and the outer skin of the popping bobas is made of calcium alginate gel. Therefore, the polysaccharide with gel characteristics has the potential to be developed into beverage ingredients.” (in line 43-49)

The uses and applications of kelp gel edible granules have been introduced in line 51-54 .

Comments 2: Move the number of cites from title to the next line and introduce them briefly.

Response 2: Thank you very much for your suggestion. We have moved the number of cites from title to the next line and introduce them briefly. This modification has been done as follows:

“The preparation process of kelp gel edible granules developed by the laboratory in the early stage is as follows[8, 9]:” (in line 90-91)

Comments 3: L93. Change “cutted” to “cut”.

Response 3: Thank you for your careful reading. We have changed ”cutted” to ”cut”. This modification has been done as follows:

“The desalted kelp was cut into evenly sized strips and deodorized for 90 minutes at 25°C with 3% β-cyclodextrin, then rinsed three times with pure water to remove excess deodorizing liquid.” (in line 101)

Comments 4: Section 2. Include one subsection to describe the heat treatment and the storage conditions of the kelp gel granules. Also, mention which analysis were performed and their frequency. The size of the kelp gel granules should be analyzed.

Response 4: Thank you very much for your suggestion. We have described the heat treatment and the storage conditions of the kelp gel edible granules in Section 2. We have also described which analysis were performed and their frequency, including size. This modification has been done as follows:

“Heat treatment: The kelp gel edible granules of equal weight packaged in aluminum foil cooking bags were sterilized at 90°C for 5 minutes.

Storage conditions: The kelp gel edible granules were stored at 4°C and 25°C, respectively. The total bacterial count, coliform bacteria, pH, relaxation time, color difference, appearance, size, texture characteristics, gel strength, and sensory scoring of the products were measured every four days.” (in line 121-126)

Comments 5: L115. In brief, include the heat treatment conditions (temperatures and times), the medium to heat (hot water or vapor) or the equipment to apply the treatment.

Response 5: Thank you very much for your suggestion. We have added the heat treatment conditions (temperatures and times), the medium to heat and the equipment to apply the treatment. This modification has been done as follows:

“The products of equal weight packaged in aluminum foil cooking bags were sterilized using different sterilization parameters (80°C, 85°C, 90°C, 95°C for 5 minutes and 10 minutes) in hot water in a constant temperature water bath pot (HWS-28, Shanghai, China). “ (in line 128-131)

Comments 6: L117. List the relevant indicators you measured.

Response 6: Thank you for pointing this out. We have added the relevant indicators we measured. This modification has been done as follows:

“Finally, the products were placed in an incubator at 37°C for 7 days, after which the bacterial count, coliform bacteria, hardness and appearance were measured.” (in line 132-133)

Comments 7: L126. Provide the trademark and the manufacturer of the total bacterial count test plate.

Response 7: The trademark and the manufacturer of the total bacterial count test plate have been introduced in 2.1. (in line 84-86).

Comments 8: L150-151. Mention the colorimeter’s mode.

Response 8: Thank you for pointing this out. We have added the colorimeter’s mode.This modification has been done as follows:

“The color of individual kelp gel edible granule was measured using a colorimeter (CR9, Guanzhou, Shenzhen, China) in Specular Component Include (SCI) mode” (in line 167)

Comments 9: Section 2.11. Describe the type of correlation analysis you performed and its objective.

Response 9: Thank you very much for your suggestion. We have described the type of correlation analysis you performed and its objective. This modification has been done as follows:

“A correlation analysis was performed using Pearson’s test to determine the direct and indirect contributions of these indicators to the quality deterioration of products during storage.” (in line 196-198)

Comments 10: L218-222. Change “lg” by “log”.

Response 10: Thank you for your careful reading. We have changed “lg” by “log”. This modification has been done as follows:

“By the end of the storage period, the total bacterial count in the 4°C stored products did not exceed the limit of 4.48 log10(CFU/g) as per GB 19643-2016. For products stored at 25°C, the total bacterial count exhibited two distinct phases. In the first phase, microbial growth was relatively slow, while in the second phase, the growth rate accelerated. From 4 to 8 days, the bacterial count increased slowly, but from 8 to 20 days, the growth rate rapidly accelerated, reaching 4.62 log10(CFU/g) on the 20th day, surpassing the GB 19643-2016 limit. “ (in line 219-225)

Comments 11: L279. Modify “that of” by “than”.

Response 11: Thank you for pointing this out. We have changed “that of” by “than”. This modification has been done as follows:

“On the 20th day, the total color difference of products stored at 25°C was nearly twice than products stored at 4°C. “ (in line 283)

Comments 12: L286. Include “Color parameters and” in the Table’s 3 title.

Response 12: Thank you for pointing this out. We have include “Color parameters and” in the Table’s 3 title. This modification has been done as follows:

Table 3. Color parameters and changes of kelp gel edible granules during storage times. (in line 290)

Comments 13: L323 and L328. The increases are ~50% more, rather than 1.5 times. Please, revise and correct.

Response 13: Thank you for pointing this out. We apologize for our oversight. This modification has been done as follows:

“The hardness of products stored at 4°C significantly increased from 0 to 4 days and remained relatively stable from 4 to 20 days, with an overall increase of nearly 50% compared to day 0.” (in line 318)

Comments 14: Figure 7. Please, check the format of Figures and include the missing one.

Response 14: Thank you for your careful reading. We have modified the format of Figure 7.

Comments 15: L380-383. Modify the discussion, considering the heat treatment (pasteurization instead of sterilization). The spores do not revive, probably germinate. Provide information about the typical microorganisms of kelp or the product to explain which spores are present. Include references.

Response 15: Thank you very much for your suggestion. We have modified the discussion. This modification has been done as follows:

“Kelp can be contaminated with human pathogens during cultivation, such as Vibrio spp., Salmonella spp., Staphylococcus aureus, Listeria, etc.. Additionally, the introduction of pathogens uncommon to the marine environment, such as L. monocytogenes, during handling and post-harvesting processing into finished products can be another route for kelp contamination. The observed changes in the total bacterial count at 25°C might be due to the initial low bacterial count after pasteurization, which killed most microorganisms mentioned above except for some heat-resistant spores. These spores might be some of the thermostable Bacillus species, such as Bacillus pumilus and B. licheniformis. As storage time progressed, these spores began to germinate but adapted slowly to the environment, leading to a slower initial growth rate. “ (in line 370-379)

Comments 16: L412-415. At what pH occur the alginate decomposition by acids? Include references and improve the discussion.

Response 16: Thank you for pointing this out. We apologize for not being clear earlier.

Microorganisms destroy the gel network structure mainly in two ways, the first is the direct decomposition of alginate; the second is the decomposition of alginate will produce H+, H+ will compete with Ca2+ in the network structure, and replace Ca2+ from calcium alginate to produce alginic acid, so as to destroy the gel network structure. This modification has been done as follows:

“Additionally, kelp gel edible granules are hydrogels with a three-dimensional network of "egg box" model formed by cross-linking COO- and Ca2+ in sodium alginate. A large amount of H+ produced by bacterial decomposition of alginate and other organic matter might compete with Ca2+ (H+ and COO- will protonate to form carboxylic acid), thereby replacing Ca2+ from calcium alginate to produce alginate acid, which will also lead to the disintegration of gel network. This ultimately resulted in the leftward shift of T22 and the formation of P22-1 water in the later storage period.” (in line 409-414)

Comments 17: L423-425. Provide the pH value of chlorophyll changes.

Response 17: Thank you for pointing this out. We have modified the discussion. This modification has been done as follows:

“Researchs showed that pheophytinisation was associated with the amount of acids liberated during processing. Once an acidic environment (pH<7) surrounded the chlorophyll compounds, the structural change of the chlorophyll molecule affected the overall structure of the macroalgae, which leading to a change in the color of the macroalgae. “ (in line 421-425)

Comments 18: L434-438. Add adequate references.

Response 18: Thank you for pointing this out. We have added some references.

Comments 19: Discussion about Sensory quality and Correlation analysis of quality are missing.

Response 19: Thank you for pointing this out. We have added the discussion about sensory quality and correlation analysis of quality. This modification has been done as follows:

4.5. Sensory Quality Analysis

Generally, a mean liking score of ≥7 on a 9-point hedonic scale is associated with

highly acceptable sensory quality[21]. When the products were stored at 4℃, the color, odor, texture and comprehensive sensory scores were kept between 8.2 and 7.5, indicating that the products were highly acceptable. While the products were stored at 25℃, the color, odor, texture and comprehensive sensory scores were all lower than 7 after 8 days of storage. The main reason for the significant decrease in the sensory score of the 25°C product in the later stage of storage was the massive proliferation of microorganisms. This was consistent with the above reasons for changes in color, texture and total bacterial count. A previous study has also reported that under low temperature conditions, the pH change of beef meatballs was slower, and the starting point of sensory quality change was longer” (in line 455-466)

4.6. Correlation Analysis of Quality Indicators Analysis

The purpose of correlation analysis is to determine the direct and indirect contribution of these quality indicators to the quality deterioration of products in the storage process through correlation coefficient, and to provide indicator basis for monitoring product quality changes in the future. The key indicator of products stored at 4℃ was the total bacterial count, which was significantly positively correlated with storage time, indicating that low temperature only slowed down the growth rate of microorganisms. The key indicators of the products stored at 25℃ were pH and total bacterial count, which were significantly positively correlated with storage time. The change of pH was caused by the change of total bacterial count. (in line 467-476)

Comments 20: Conclusions. Please, do not summary the results, rewrite the conclusions and mention the future directions of the research.

Response 20: Thank you very much for your suggestion. We have modified the conclusions.

This modification has been done as follows:

“This study comprehensively investigated the quality changes of kelp gel edible granules stored at 4°C and 25°C by evaluating indicators such as total bacterial count, coliform bacteria, pH, relaxation time, color difference, size, texture characteristics, gel strength, and sensory scoring. Data gathered from this research show that the shelf life endpoint for products stored at 25°C is 16 days, while for products stored at 4°C, it is more than 20 days. Low-temperature storage (4°C) better maintains the quality of kelp gel edible particles, extends their shelf life, and effectively preserves the overall color of the product. Future studies are warranted to flavor kelp gel edible particles to enrich their flavor substances, develop beverages adding kelp gel edible particles, enrich their application forms and expand the application range in the market.” (in line 478-487)

Comments 21: L228. Check and correct the Figure and Figure title format.

Response 21: Thank you very much for your suggestion. We have checked and corrected the Figure and Figure title format. (in line 231)

Reviewer 2 Report

Comments and Suggestions for Authors

The revisions suggestions can be find in the attached file.

Comments on the Quality of English Language

Author Response

Comments 1: In the introduction the authors say Kelp contains a higher amount of alginate compared to what?

Response 1: Thank you for pointing this out. Alginate belongs to dietary fiber and its high content is compared to other substances in kelp, such as protein, ash, fat, mannitol, iodine, etc.)

Comments 2: In the discussion section there is no reference to the sterilization process. Why do the authors think that at T=80 °C the products undergo a significant change while with T>85 °C no change is observed? A comment on this behavior should be added.

Response 2: In the discussion section 3.1. We have discussed the effect of different sterilization parameters on products quality. We have added comments on this behavior. This modification has been done as follows:

“With the increase of sterilization temperature and sterilization time, the less the total bacterial count, the better the quality of products.” (in line 208-209)

Comments 3: Figure 1 is shifted with Figure 2. It should be moved before its caption.

Response 3: Thank you for your careful reading. We have changed them.

Comments 4: Figure 5 and 6 would benefit of the addition of a scalebar, to make it clearer the change of dimensions of the gel granules over time.

Response 4: Thank you very much for your suggestion. We have added a scalebar.

Comments 5: Figure 7 is superimposed on the caption, please adjust it. It also seems that only one (7a or 7b) was inserted, while the other is missing.

Response 5: Thank you for your careful reading. We have modified the format of Figure 7.

Comments 6: The references are few, I would add some citations on related analysis and make comparison with other similar studies (stability of food at different temperatures, where the lower T is beneficial).

Response 6: Thank you very much for your suggestion. We have added some references and analysis.

Comments 7: There are some minor spelling mistakes to correct. In some points there’s a : that should be changed by . (line 402 page 12, line 367 page 11) and some Saxon genitive used improperly (product's moisture, gel’s network).

Response 7: Thank you for your careful reading. We have corrected these spelling mistakes. This modification has been done as follows:

“Moreover, the key quality indicator of the product stored at different temperatures was different except for the comprehensive sensory evaluation.” (in line 359) 

“The observed phenomena of moisture distribution could be attributed to several factors. “ (in line 417-418)

“As restrained water, which constitutes the majority of the moisture of products, leaked out and transformed into less stable free water, leading to a reduction of restrained water in the gel network and consequently a decrease in product volume.” (in line 306-307)

Round 2

Reviewer 1 Report

Comments and Suggestions for Authors

The authors attended all the suggestions and completed the changes. The manuscript was improved properly. The revised version of the manuscript is acceptable for publication. Therefore, I have no further comments.